# Incidence of SARS-CoV-2 infection among healthcare workers before and after COVID-19 vaccination in a tertiary paediatric hospital in Warsaw: A retrospective cohort study

Beata Kasztelewicz[1]*, Katarzyna Skrok[1], Julia Burzyńska[1], Marek Migdał[2], Katarzyna Dzierżanowska-Fangrat[1]

1 Department of Clinical Microbiology and Immunology, The Children's Memorial Health Institute, Warsaw, Poland, 2 CEO of the Children's Memorial Health Institute, Warsaw, Poland

* b.kasztelewicz@ipczd.pl

**Data Availability Statement:** All files are available from the Mendeley Data database database accession: DOI:

## Abstract

A retrospective observational study was conducted among healthcare workers (HCWs) in a tertiary paediatric hospital. The study covered the period before and after implementation of the vaccination programme and evaluated the incidence of new SARS-CoV-2 infections in both periods. Risk factors of the new SARS-CoV-2 infection and COVID-19 vaccine effectiveness was also assessed in a real-world setting. The overall incidence of SARS-CoV-2 infections among HCWs in the study period was 19.4% with a high proportion of asymptomatic individuals (45.1%). The incidence before vaccination was 16.6% and nurses had a higher risk of infection, while physicians had a reduced risk (OR 1.80, 95% CI 1.29–2.52; and OR 0.45, 95% CI 0.30–0.68). Within two months of implementation, the programme achieved a high (88.9%) vaccination coverage in our cohort, although some disparities in vaccination rates were observed. In particular, older individuals, physicians, those working in clinical settings, and those previously uninfected were more likely to be vaccinated. The overall incidence of SARS-CoV-2 infection after vaccination deployment was 6.4% (40.0% in unvaccinated individuals and 3.2% in individuals vaccinated with at least one dose). The estimated vaccine efficacy was high (95.0%) in fully vaccinated HCWs and similar to those observed previously in clinical trials and real-world settings.

## Introduction

The ongoing pandemic of the coronavirus disease 2019 (COVID-19) caused by the novel severe acute respiratory syndrome coronavirus 2 (SARS-CoV-2) has had significant impact worldwide. Healthcare workers (HCWs) are considered to be at high risk of exposure to SARS-CoV-2 both in the community and in the workplace and may play a critical role,

10.17632/4t2z4k2m5z.1 URL ID: https://data.mendeley.com/datasets/4t2z4k2m5z

**Funding:** The author(s) received no specific funding for this work.

**Competing interests:** The authors have declared that no competing interests exist.

especially when asymptomatic, in the transmission of the infection within the workplace (both to their patients and co-workers) as well as in the community [1].

Preventing SARS-CoV-2 infection among HCWs is critical to ensure safety for both patients and HCWs, to contain the ongoing pandemic, and to maintain a functioning healthcare system. Many preventive measures have been implemented throughout the pandemic to contain the spread of the SARS-CoV-2 infection in healthcare settings. Notably, there has been a paradigm shift in the infection control practices against respiratory infections, which includes widespread testing of patients and HCWs, including asymptomatic individuals [2, 3]. Universal testing of HCWs enabled the prompt identification of asymptomatic and presymptomatic individuals, determined the effectiveness of control measures, and helped prevent transmission to patients and co-workers [1, 4, 5]. Furthermore, COVID-19 vaccines were developed to overcome the pandemic.

In Poland, COVID-19 vaccination programme started on December 27, 2020, using the Pfizer-BioNTech BNT162b2 vaccine. HCWs were among the initial groups that prioritized for vaccination. The introduction of vaccination in Poland coincided with the descending phase of the second wave of the pandemic. At the end of December 2020, 1,289,293 COVID-19 cases were reported in Poland. Masovian Voivodeship was the region most affected by COVID-19, with Warsaw having the highest number of infections, with 64,813 confirmed cases by the end of 2020 [6, 7]. Between February and April 2021, Masovian Voivodeship, like the rest of Poland, experienced a third wave of pandemics, with a peak of 5,264 daily cases reached on March 26, 2021. The number of cases decreased by June 2021, and by the end of August 2021, there were 45 daily cases of SARS-CoV-2 infections in Masovian Voivodeship [7, 8].

An early report on the global burden of the COVID-19 pandemic on HCWs showed high variability in HCW infection percentage among total cases across regions, with a median of 10.04% and range from 0 to 24.09%. Of note, Poland was one of the countries with high percentage of HCWs infections accounting for 17.07% of cases [9].

There is limited data on the incidence of SARS-CoV-2 infection among HCWs beyond the second wave of the pandemic, and limited data on the incidence of infection before and after the implementation of the vaccination programme, especially for European countries [10]. So far, most studies performed in Poland were voluntary seroprevalence studies, so the representativeness of the results obtained may be limited [11, 12]. Knowing the factors associated with SARS-CoV-2 infection in HCWs informs preventive measures and improves the protection of HCWs and patients.

To further explore factors associated with SARS-CoV-2 infection, we retrospectively analysed the real-world testing database obtained between October 20, 2020, and August 31, 2021, at The Children's Memorial Health Institute (CMHI), a tertiary paediatric hospital in Warsaw. In this setting, 2,332 HCWs participated in the universal screening programme for early identification of the SARS-CoV-2 infection. The aim of the study was to analyse the incidence of new SARS-CoV-2 infections among HCWs (before and after vaccination with BNT162b2) and to explore the demographics and work-related factors associated with SARS-CoV-2 infection. We also attempted to assess the vaccine effectiveness in a real-world setting.

## Materials and methods

### Study design

This was a cohort, retrospective study with secondary data, conducted with the Strengthening the Reporting of Observational Studies in Epidemiology (STROBE) guidelines for reporting observational studies [13], performed among HCWs at the CMHI in Warsaw (S1 File)

**Setting.** The CMHI is the largest tertiary paediatric hospital and research institute in Poland, with nearly 600 beds and approximately 2,430 HCWs, including employees and trainees. As previously described [14], beginning on March 9, 2020, a set of infection prevention and control measures were implemented at the CMHI to contain the spread of SARS-CoV-2 within the hospital. The SARS-CoV-2 RT-PCR testing for symptomatic personnel, coupled with contact tracing, was implemented starting from March 17, 2020. In addition, since October 20, 2020 (the beginning of the second wave of the COVID-19 epidemic in Poland), a universal SARS-CoV-2 screening programme was deployed at the CMHI for all HCWs using a RT-PCR assay of nasopharyngeal swabs. Up to May 6, 2021 (corresponding to the end of the third pandemic wave), there were seven rounds of screening which made it possible to identify and isolate asymptomatic and presymptomatic HCWs. The HCW vaccination programme with the Pfizer-BioNTech BNT162b2 vaccine began on January 4, 2021, and up to February 28, 2021, approximately 80% of HCWs received at least one dose of the vaccine.

**SARS-CoV-2 RNA testing of HCWs.** RT-PCR tests of combined nasal and oropharyngeal swab samples were performed at the Department of Clinical Microbiology and Immunology using initially the MutaPLEX® Coronavirus (SARS-CoV-2) real-time RT-PCR Kit (ImmunDiagnostik AG, Bensheim, Germany), switching to the SARS-CoV-2 Triplex PCR Kit (Astra Biotech, Berlin, Germany; from November 2020). The capacity was approximately 300 tests per day. Results of the SARS-CoV-2 RNA testing were communicated to HCWs within 8 hours by text message. All positive results were reported to the Infection Control and Prevention (ICP) unit on a daily basis. Daily trends in the positive result of HCW testing were monitored to identify areas of concern to be targeted for reactive approach testing. The positive results of the tests performed in the community were verbally reported to the ICP unit by the HCWs and were also entered into the database, while the negative results were not. In the case of positive PCR test results, HCWs were required to self-isolate for at least 10 days.

## Participants, data collection and management

All HCWs (clinical and non-clinical) who had at least one PCR test performed as part of the universal screening were included in the study.

Demographic, occupational, epidemiological (if applicable), and laboratory data were collected for each participant up to August 31, 2021. The database aggregates data from multiple sources. In particular, the following data were extracted directly from the prospectively maintained laboratory records: demographic (age and gender); results of the SARS-CoV-2 RNA RT-PCR test from longitudinal universal screening (between October 20, 2020, and May 6, 2021, which corresponds to epidemiological (epi) week 43, 2020 and epi week 18, 2021, the period covering the second and third epidemic waves in Poland); SARS-CoV-2 RNA results from contact tracing, symptomatic testing or testing upon return to work after sick leave (due to respiratory illness) collected from October 20, 2020, up to August 31, 2021 (epi week 43, 2020, to epi week 35, 2021); IgG antibodies to the nucleocapsid protein of SARS-CoV-2 (if available). An invalid PCR test for any reason was excluded from the analysis. Data on profession and area of work were collected from the human resources database. Data on vaccination status (including vaccine type, number of doses received, and vaccination date) were extracted from the database maintained by the Clinical Work Support Section (a unit which coordinated HCW vaccination at the CMHI). In addition, in the case of SARS-CoV-2 RNA positive individuals, data from the ICP records were retrieved regarding self-reported symptoms, presumed source of infection, contacts and history of exposure, as well as the date of the positive SARS-CoV-2 RT-PCR test (if performed in the community). Symptomatic SARS-CoV-2 infection was defined as PCR confirmed infection with one or more of the symptoms suggesting

COVID-19 (including fever, cough, shortness of breath, anosmia, ageusia, myalgia, upper respiratory symptoms, diarrhea, vomiting, severe fatigue) 14 days before to seven days after the first PCR positive test.

All data were analyzed anonymously in April 2023.

Fig 1 illustrates the study timeline. The study population and definition of the follow-up period for the incidence of SARS-CoV-2 infection in HCWs before vaccination (study phase 1) and after vaccination (study phase 2) are presented in Table 1.

## Statistical plan

**Outcome measures.** The main outcomes were the incidence of new SARS-CoV-2 infection in HCWs before and after vaccination, and the vaccine effectiveness (VE) against any SARS-CoV-2 infection. The outcomes were analysed for predefined subgroups according to age, gender, profession, workplace, and vaccination status.

**Sample size.** The sample size was calculated based on the previously published prevalence of SARS-CoV-2 infection of 1.3% among fully vaccinated and 10.1% among unvaccinated HCWs [15]. Considering an alpha error of 0.05 and a study power of 80%, it is necessary to include at least 102 and 120 HCWs in the fully vaccinated and unvaccinated groups, respectively.

**Sensitivity analysis.** Sensitivity analysis was performed to examine the robustness of the results. We performed complete case analysis (i.e. after inclusion of previously infected HCWs, those with missing vaccination status, or no follow-up data) to cover more possible events and to investigate influences of missing data on the study results.

**Statistical analysis.** Descriptive statistics were used to characterise the study cohort. Continuous variables were summarised as median and interquartile ranges (IQR). Categorized variables were summarized using percentages and counts. The associations between variables were tested using the Chi-square test for categorical variables and the Mann–Whitney U test for

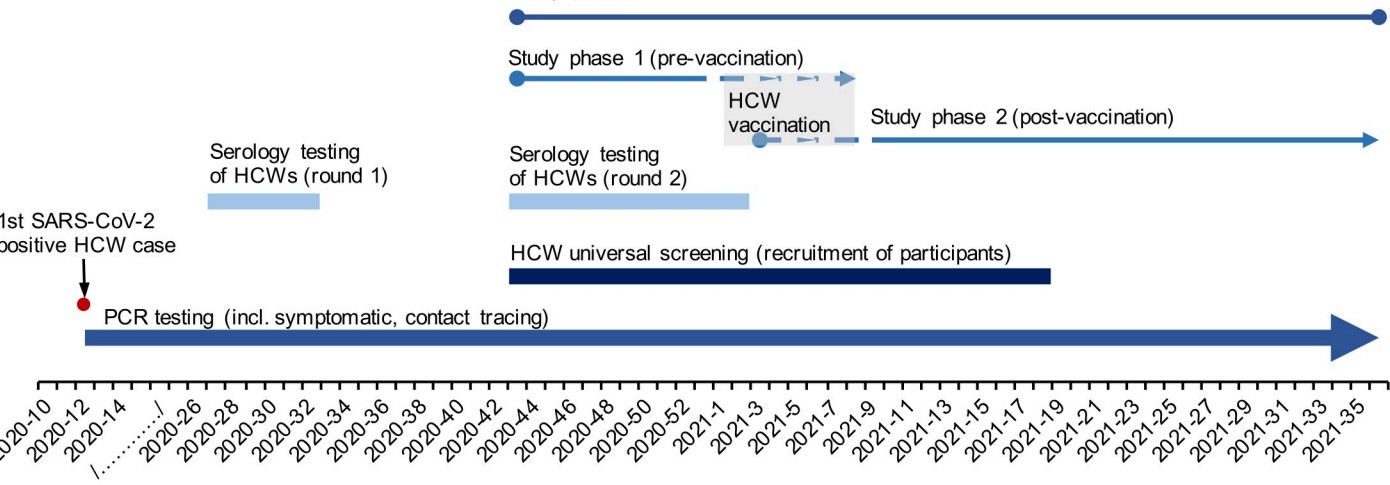

**Fig 1. Study timeline.** The study cohort involving healthcare workers (HCWs) at The Children's Memorial Health Institute (CMHI) in Warsaw. Study participants were recruited between October 20, 2020, and May 6, 2021 (corresponding to the period of HCWs universal screening implemented at the CMHI between epidemiological week 43, 2020, and week 18, 2021). The COVID-19 vaccination programme began on January 4, 2021, and HCWs who received at least 1 dose of the vaccine up to February 28, 2021, were considered vaccinated. The study was divided into phase 1 (pre-vaccination) and phase 2 (post-vaccination). Demographic, occupational, laboratory and epidemiological data were collected retrospectively spanning the period between October 20, 2020 and August 31, 2021.

**Table 1. Study populations and definitions of follow-up periods used for the incidence analysis.**

| Study phase: | Description | Start of follow-up period | End of follow-up period (whichever occurred earlier): |
|---|---|---|---|
| 1: before vaccination | SARS-CoV-2 uninfected[a] | October 20, 2020 (epi week 43) | First PCR-positive test result |
| | | | The receipt of vaccine dose 1 February 28, 2021 (epi week 8) |
| 2: after vaccination | **Unvaccinated** | March 1, 2021 (epi week 17) | The receipt of vaccine dose 1 |
| | SARS-CoV-2 uninfected[a] no vaccine dose or <14 days after dose 1 | | First PCR-positive test result |
| | | | August 31, 2021 (epi week 35) |
| | **partially vaccinated** SARS-CoV-2 uninfected[a], ≥14 days after dose 1 and < 14 days after dose 2 | ≥14 days after dose 1[b] | The receipt of vaccine dose 2 |
| | | | First PCR-positive test result |
| | | | August 31, 2021 (epi week 35) |
| | **fully vaccinated** SARS-CoV-2 uninfected[a] | ≥14 days after dose 2[b] | First PCR-positive test result |
| | and ≥14 days after dose 2 | | August 31, 2021 (epi week 35) |

[a] based on negative PCR test results and/or negative IgG anti-nucleocapsid

[b] for the incidence analysis we used a wash-out period of 0–13 days after each dose of vaccine and adopted ECDC's criteria to determine vaccination status at the time of PCR testing (source: European Centre for Disease Prevention and Control. *Interim analysis of COVID-19 vaccine effectiveness in healthcare workers, an ECDC multi-country study, May 2021- July 2022. ECDC: Stockholm;2022*)

continuous ones. Incidence of new cases of SARS-CoV-2 infection per 1,000 HCWs at risk was calculated for each epi week. The denominator includes HCWs at risk in a specific epi-week. The numerator is the number of new positive SARS-CoV-2 cases among the HCWs. We stratified primary SARS-CoV-2 infection rates by demographic characteristics. Similarly, we grouped occupational categories to ensure groups previously noted to be at higher risk of infection were represented separately (e.g. including nurses, physicians, and others with and without direct patient contact). We applied previously described occupational subgroup criteria [14].

Univariable and multivariable logistic regression models were used to assess factors associated with the primary outcome of SARS-CoV-2 infection. Explanatory variables for multiple regression analysis were selected based on association with SARS-CoV-2 infection in univariate analysis and adjusted for age, gender, and number of PCR tests per person. Age and gender were applied as confounders based on previously documented associations with SARS-CoV-2 infection among HCWs [16]. We also adjusted for the number of PCR tests per person, as infected HCWs in our cohort had lower number of tests performed than those uninfected during both study phase 1 and 2.

Similarly, adjusted odd ratios (aORs) for SARS-CoV-2 infection in vaccinated HCWs as compared with unvaccinated HCWs were estimated using logistic regression models to determine vaccine effectiveness (VE) against any infection. VE was calculated using the following equation VE = (1 –adjusted OR) x 100% with 95% CI; separately for partially and fully vaccinated HCWs.

Furthermore, VE was estimated by age category (< 50 years vs. 50+ years) and professional group for fully vaccinated HCWs. The number of cases in the partially vaccinated group precludes the estimation of VE in subgroups. We did not evaluate the effect of circulated variants on SARS-CoV-2 incidence on VE as all of the SARS-CoV-2 infections occurred within the pre-Delta period, with dominance of the Alpha (B.1.1.7) variant. According to ECDC data, the Delta Variant (B.1.617.2) became the dominant variant in Poland in epi week 26, 2021 [17]. We did not test heterogenicity by vaccine type as within the first two months of the programme all HCWs were vaccinated with the Pfizer-BioNtech BNT162b2 vaccine.

The data set was extracted and reviewed in Microsoft Excel. Sample size calculation and statistical analyses were performed using the Statistica data analysis software system (TIBCO Software Inc.), version 13. Two-side tests with a p-value of $< 0.05$ were considered statistically significant.

### Ethical considerations

The study involving humans was reviewed and approved by the Bioethics Committee at the Children's Memorial Health Institute in Warsaw (Ref. no. 42/KBE/2020 with later revision on March 31, 2021), and granted a waiver of consent since the data were retrospective and were anonymized prior to review, and statistical analysis. Data were accessed between April, 30 2021 and September, 1 2022. During the data collection, all personal data were deidentified. Data were analyzed anonymously and reported only in aggregated form, to further ensure confidentiality of data.

## Results

### Characteristics of the study population

Of the 2,332 HCWs who participated in the universal screening programme, 1,967 (84.3%) were female. The median age was 46.9 (IQR: 36.4–55.4) years. Approximately one third (32%) were nurses. One hundred and eighty-four (7.9%) HCWs worked in the COVID-19 area (Table 2).

Beginning on March 17, 2020 (the first case of a SARS-CoV-2 infected HCW identified at the CMHI) and up to October 19, 2020 (the start of universal screening of HCWs), we identified 86 HCWs with SARS-CoV-2 infection from the 2,332 HCWs, which corresponds to a baseline prevalence of 3.7%.

### SARS-CoV-2 RNA PCR testing

In total, 11,797 samples from 2,332 HCWs were tested as part of the universal screening programme between October 20, 2020, to May 6, 2021. After excluding 199 (1.7%) invalid test results, the median number of tests per person was 6 (IQR: 3–7). In addition, 1,298 PCR test results performed outside the screening and up to August 31, 2021, were available for 934 (40.1%) of the 2,332 HCWs (median 1 sample per person, IQR: 1–2).

Overall, 457 (19.6%) of the 2,332 HCWs had positive SARS-CoV-2 PCR test results within the study period, including 452 new infections (19.4%) and 8 (0.2%) reinfections. Almost half (45.1%) of the infections were asymptomatic and 48.6% were identified during the universal screening programme. Among 281 HCWs with laboratory confirmed SARS-CoV-2 infection, for whom data on the probable source of infection were available, the most common source of infection was the community (44.1%), followed by the household (32.7%), while the rate of infection at the workplace was only 14.2%.

Fig 2 shows the incidence of SARS-CoV-2 infection among HCWs at the CMHI and in the general population in the Mazovian voivodeship and S1 Fig reports the new weekly cases of SARS-CoV-2 infection among HCWs at the CMHI by testing mode. The weekly incidence of new cases among HCWs at the CMHI fluctuated during the study period, corresponding to the dynamic of SARS-CoV-2 transmission in the community. In particular, most of the cases were identified between epidemiological (epi) week 43 and 49, 2020, which encompassed the peak of the second epidemic wave in Poland. The incidences among the HCWs decreased during the first three months of the vaccination programme and then increased again and peaked at epi week 13, 2021, which corresponded to the peak of the third wave in Poland, caused by

**Table 2. Characteristics of 2,332 HCWs participating in the universal screening programme (SARS-CoV-2 PCR testing).**

| Characteristics | Total, n = 2332 |
|---|---|
| Age, median (IQR), years: | 46.9 (36.4–55.4) |
| Female gender, n (%): | 1967 (84.3) |
| Professional category[a], n (%): | |
| nurse | 746 (32.0) |
| physician | 506 (21.7) |
| other with direct patient contact | 313 (13.4) |
| other without direct patient contact | 767 (32.9) |
| Clinical department, n (%) [b]: | 1675 (71.8) |
| Working in COVID-19 area, n (%): | 184 (7.9) |
| Wards, n (%): | |
| medical | 975 (41.8) |
| surgical | 189 (8.1) |
| intensive care | 127 (5.4) |
| auxiliary | 252 (10.8) |
| ambulatory | 132 (5.7) |
| laboratory | 144 (6.2) |
| maintenance | 98 (4.2) |
| administration | 362 (15.5) |
| other | 53 (2.3) |
| Previous SARS-CoV-2 infection, n (%)[c]: | 86 (3.7) |
| Total no. of HCWs who tested SARS-CoV-2 positive during the study, n (%)[d]: | 457 (19.6) |
| SARS-CoV-2- positive cases identified by screening testing[e]: | 222 (48.6) |
| SARS-CoV-2- positive cases identified outside the screening[f]: | 235 (51.4) |
| Symptoms at the time of positive PCR, n (%): | |
| symptomatic [g] | 241 (52.7) |
| asymptomatic | 206 (45.1) |
| unknown | 10 (2.2) |
| Reported source of infection, n (%): | 281 (61.5) |
| Source of infection, n (%): | |
| community | 124 (44.1) |
| household | 92 (32.7) |
| workplace | 65 (14.2) |

Abbreviation: IQR–interquartile range

[a] *other with direct patient contact* includes: patient care technician, physical therapist, radiation therapist, psychologist, medical assistant, audiologist, pedagogue, speech pathologist, clinic engineer, medical technician, dental assistant, anthropologist; *other without direct patient contact* include: office worker, secretary, laboratory worker, kitchen worker, pharmacist, dietician, medical sterilization technician, driver, labourer, store person, IT worker, public health worker, manager, security officer, chaplain, parking attendant

[b] clinical departments include: medical, surgical, auxiliary medical, ambulatory, intensive care; nonclinical departments include: administration, laboratory, maintenance, pharmacy

[c] 73 HCWs tested positive by PCR and 13 were positive for SARS-CoV-2 by nucleocapsid IgG before the screening

[d] there were 460 PCR positive tests in 457 HCWs, including 452 new infections and 8 reinfections (3 HCWs had both a new infection and a reinfection detected within the study period)

[e] in 11 HCWs the universal screening tests coincided with a return to work after sick leave due to respiratory illness

[f] 235 HCW with symptoms or in contact with a case of COVID-19 tested within the period that covered the universal screening programme, only one HCW was found positive afterward

[g] symptoms fourteen days before to seven days after the first positive PCR test. Four out of 241 HCWs required hospitalisation, one died

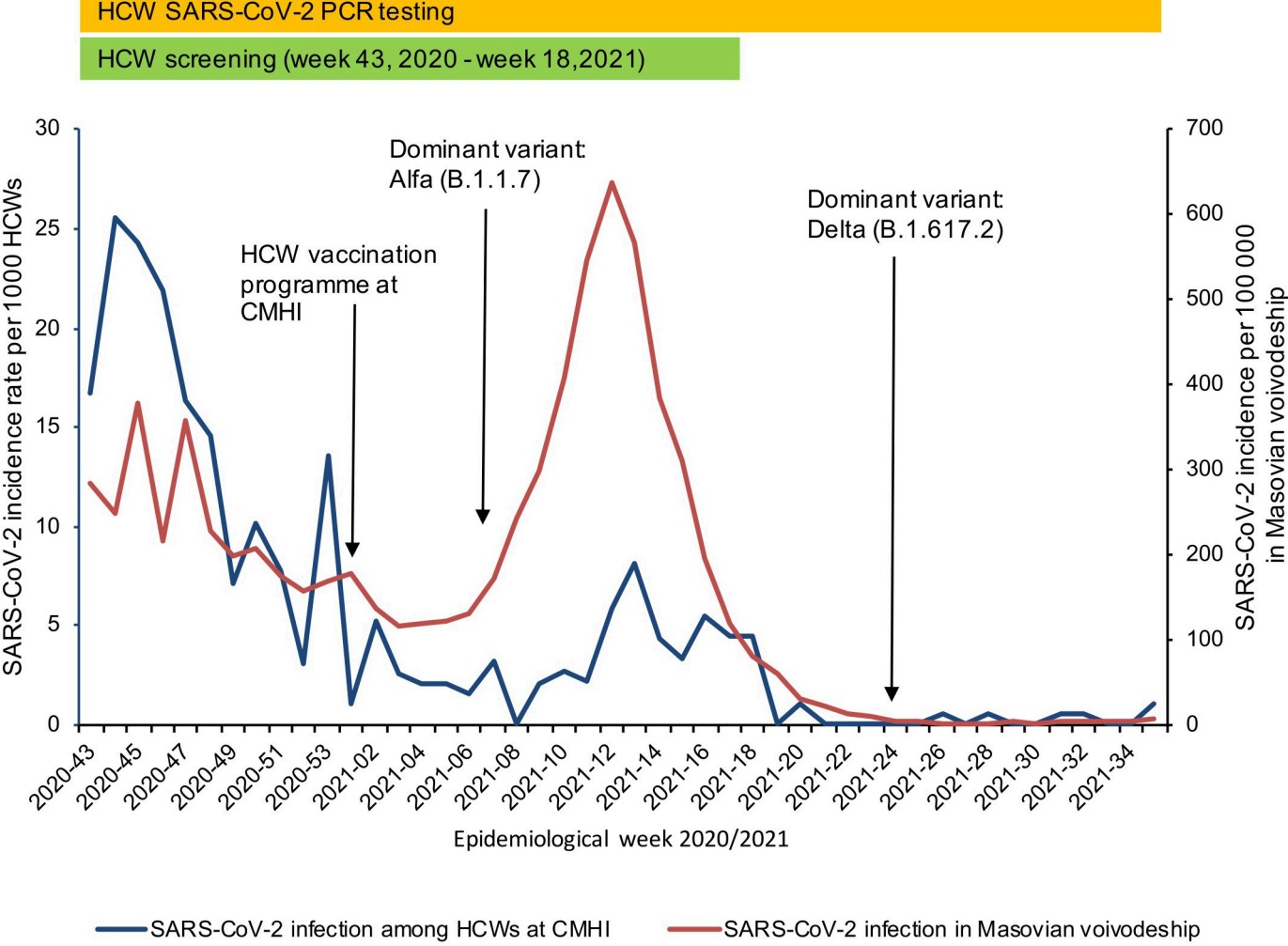

**Fig 2. Incidence of SARS-CoV-2 infections among HCWs at the CMHI and in the general population of the Mazovian voivodeship.**

variant Alpha (B.1.1.7; S2 Fig). This later peak of incidence was relatively lower among HCWs than in the community.

## Study phase 1: Incidence of new SARS-CoV-2 infection acquired before vaccination rollout

After exclusion of 185 HCWs with previous SARS-CoV-2 infection (n = 86) and those with the first PCR test performed after the implementation of the vaccination programme (n = 99), the study phase 1 consisted of 2,147 HCWs (Fig 3; Phase 1 Study flow chart). During the 18-week period, 357 HCWs acquired SARS-CoV-2 infection, which corresponds to a cumulative incidence of 16.6%.

More than half of the infected HCWs were asymptomatic (51%). The HCWs who tested positive for SARS-CoV-2 were similar in age and gender. The lowest percentage of SARS-CoV-2 infections was found among physicians (10%) and the highest among nurses (20%; p < 0.001). There was no significant difference in infection rates among HCWs working in wards involved and not involved in the treatment of COVID-19 patients or clinical and non-

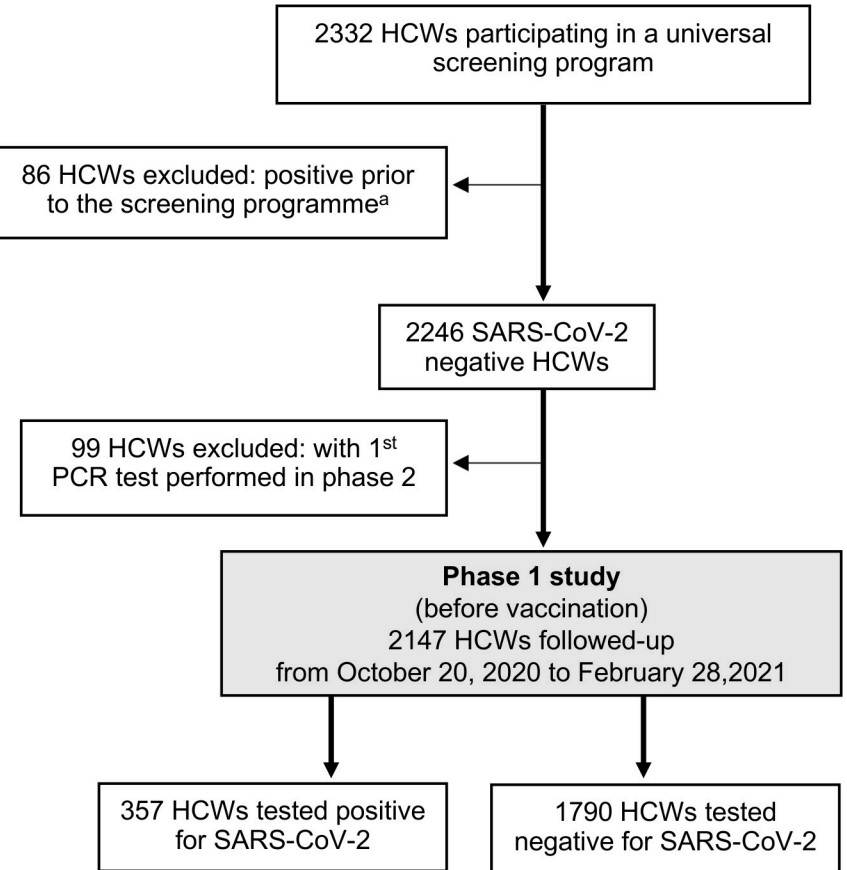

**Fig 3. Flow diagram of the participant inclusion and exclusion criteria used to define study cohort in phase 1 of the study.** [a] including 73 with SARS-CoV-2 PCR confirmed and 13 seropositive prior to the screening programme; 4 out of 86 HCWs experienced reinfection within phase 1.

clinical areas, as well as across wards (Table 3). Each HCW had 4 PCR tests performed as a median (IQR: 3–5 tests), with a higher number of tests performed for uninfected HCWs (p < 0.001).

The results of the sensitivity analysis after including 86 HCWs with previous SARS-CoV-2 infection were similar to the main analysis for all variables considered (S1 Table, n = 2,233).

Following adjustment for age, gender, and number of tests per person, there remained significant differences in the incidence of new SARS-CoV-2 infection across professional categories. Most notably, the odds of being infected were greater for nurses and lesser for physicians relative to the 'other without patient contact' category (OR 1.80, 95% CI 1.29–2.52, p = 0.001 and OR 0.45, 95% CI 0.30–0.68, p < 0.001, respectively; S2 Table).

## Vaccine coverage among HCWs

One thousand nine hundred and sixty (84.0%) HCWs of the 2,332 HCWs had known vaccination status, among them 1,743 (88.9%) HCWs had received at least one dose of the BNT162b2 vaccine by the end of February, 2021 (1,717 received two doses with a median time of 20 days between dose 1 and 2). The characteristics of the HCWs by vaccination status are presented in Table 4.

**Table 3. Characteristics of the HCWs by SARS-CoV-2 infection status during study phase 1 (pre-vaccination period; n = 2147).**

| Characteristics | Infected | Uninfected | Total | p-value |
|---|---|---|---|---|
| Total, n (%): | 357 (16.6) | 1790 (83.3) | 2147 | |
| Age, median (IQR), years: | 46.6 (37.1–54.8) | 47.3 (36.6–55.8) | 47.2 (36.6–55.6) | 0.503 |
| Female gender, n (%): | 311 (17.2) | 1495 (82.8) | 1806 | 0.090 |
| Professional category, n (%): | | | | < 0.001 |
| nurse | 139 (20.0) | 556 (80.0) | 695 | |
| physician | 46 (10.0) | 413 (90.0) | 459 | |
| other with direct patient contact | 52 (18.2) | 234 (81.8) | 286 | |
| other without direct patient contact | 120 (17.0) | 587 (83.0) | 707 | |
| Working in COVID-19 area, n (%): | | | | 0.321 |
| yes | 32 (19.4) | 133 (80.6) | 165 | |
| no | 325 (16.4) | 1657 (83.6) | 1982 | |
| Hospital department, n (%): | | | | 0.753 |
| clinical | 260 (16.8) | 1289 (83.2) | 1549 | |
| non-clinical | 97 (16.2) | 501 (83.8) | 598 | |
| Wards, n (%): | | | | 0.676 |
| medical | 149 (16.5) | 752 (83.5) | 901 | |
| surgical | 30 (17.4) | 142 (82.6) | 172 | |
| intensive care | 22 (19.8) | 89 (80.2) | 111 | |
| auxiliary | 34 (14.1) | 208 (85.9) | 242 | |
| ambulatory | 25 (20.3) | 98 (79.7) | 123 | |
| laboratory | 21 (17.5) | 99 (82.5) | 120 | |
| maintenance | 19 (21.1) | 71 (78.9) | 90 | |
| administration | 49 (14.5) | 288 (85.5) | 337 | |
| other | 8 (15.7) | 43 (84.3) | 51 | |
| Median no. of PCR tests per person, median (IQR) | 2 (1–3) | 4 (3–5) | 4 (3–5) | < 0.001 |

Abbreviation: IQR–interquartile range

The percentages were presented in rows to highlight the proportion of infected and uninfected HCWs for each level of the variables

Unvaccinated HCWs were significantly younger than vaccinated (median age: 45.9 vs 48.2 years; p = 0.002). Physicians were significantly more likely to be vaccinated than other professional groups (p < 0.001). The HCWs in non-clinical areas were less vaccinated than those working in clinical locations (85.7% vs 90.2%, p = 0.005). There were differences in the vaccination status between the wards, with HCWs working on the surgical wards being the most frequently vaccinated group (92.6%, p = 0.04). There were no significant differences in vaccination status between HCWs working in wards involved and not involved in the treatment of COVID-19 patients. The HCWs most likely to have received at least one dose of the vaccine were previously uninfected with SARS-CoV-2 (p < 0.001). Before the implementation of the vaccination programme, 352 HCWs (285 vaccinated and 67 unvaccinated) had laboratory confirmed SARS-CoV-2 infection. Unvaccinated HCWs had a shorter time period from previous infections when compared with vaccinated HCWs (median 20 vs 61 days; p < 0.001).

In sensitivity analysis restricted to 1,461 cases included in the study phase 2, the distributions of vaccinated and unvaccinated HCWs across different categories were similar to those previously presented for 1,960 HCWs with known vaccination status with the exception of the ward category (S3 Table). In contrast to the results of the main analysis, no statistical significance was observed in the sensitivity analysis for the category of wards, although the shares of

the HCWs follow a similar pattern, with the HCWs in the surgical, auxiliary, and medical wards being the most vaccinated.

## Study phase 2: Incidence of new SARS-CoV-2 infections after implementation of the vaccination programme

The incidence of SARS-CoV-2 infections was determined for vaccinated (partially and fully) and unvaccinated HCWs. HCWs with unknown vaccination status, previous SARS-CoV-2 infection, and/or without follow-up data were excluded from the incidence analysis (Fig 4; Phase 2 study flowchart). After applying the exclusion criteria, the incidence analysis included 1,461 HCWs with a median follow-up of 79 (IQR: 70–86) days.

In total, 93 of 1,461 (6.4%) HCWs tested positive for SARS-CoV-2 by the end of August, 2021. There was no significant difference between infected and uninfected HCWs across demographic and occupational categories (S4 Table). The cumulative incidence of SARS-CoV-2 infection was significantly higher among unvaccinated HCWs than in vaccinated HCWs (40.0% vs 3.2%; p < 0.001). When vaccination groups were considered, the cumulative incidence of new SARS-CoV-2 cases was 12.2% and 2.9% in partially and fully vaccinated HCWs, respectively (Fig 5). The median interval from the first vaccine dose to SARS-CoV-2 infection was 28 (IQR: 21–31, range: 18–73) days in partially vaccinated HCWs, while in the fully vaccinated group, the median interval between the second vaccine dose and infection was 62 (IQR: 49–76, range: 25–200) days.

In the sensitivity analysis, the association of vaccination status with SARS-CoV-2 infection remained significant after including 192 HCWs with unknown vaccination status as a separate category (p < 0.001; S5 Table).

In a multivariate logistic regression model with confirmed SARS-CoV-2 cases as a response variable, vaccination status was the only significant predictor for SARS-CoV-2 infection after adjusting for age, gender, and number of PCR tests per HCWs (S6 Table).

Vaccine effectiveness in preventing any SARS-CoV-2 infection was 79% (95% CI 46–92%) and 95% (95% CI 91–97%) in partially and fully vaccinated HCWs, respectively.

Vaccine effectiveness (VE) by age group and professional categories is presented in Table 5. VE for fully vaccinated HCWs did not vary by age (age group < 50 years: adjusted VE (aVE) = 95%, 95% CI 90–97%; 50+ years: aVE = 95%, 95% CI 87–98%) and was slightly lower for the other without direct patient contact category (aVE = 89%, 95% CI 73–96%) when compared with physicians (aVE = 96%, 95% CI 84–99%) and nurses (aVE = 98%, 95% CI 92–99%).

## Discussion

By using longitudinal data, our study provides robust data on the incidence of new SARS-CoV-2 infection among HCWs in a paediatric hospital during the pre- and post-vaccinations periods, encompassing the second and third wave of the COVID-19 pandemic in Poland. During the ten months of the study period, approximately one fifth (19.4%) of the HCWs susceptible to primary infection became infected with SARS-CoV-2. It is worth noting that almost half of the HCWs who had laboratory confirmed SARS-CoV-2 infection in our study did not report symptoms, suggesting that without the implementation of proactive universal screening a significant proportion of infection among HCWs would have remained undetected and that they would likely have continued working while unaware of their status, therefore presenting a risk of transmission to patients and co-workers. Although this study was not designed to address the issue of nosocomial transmission reduction through asymptomatic testing, we assume that the universal screening programme of HCWs applied in our hospital (as part of a

**Table 4. Characteristics of vaccinated and unvaccinated HCWs (n = 1960).**

| Characteristics | Vaccinated [a] | Unvaccinated [b] | p-value |
|---|---|---|---|
| Total, n (%) | 1743 (88.9) | 217 (11.1) | |
| Age, median (IQR), years: | 48.2 (38.0–56.6) | 45.9 (36.1–53.7) | 0.002 |
| Female gender, n (%) | 1463 (88.8) | 185 (11.2) | 0.617 |
| Professional category, n (%): | | | < 0.001 |
| nurse | 544 (88.9) | 68 (11.1) | |
| physician | 421 (95.7) | 19 (4.3) | |
| other with direct patient contact | 219 (85.6) | 37 (14.5) | |
| other without direct patient contact | 559 (85.7) | 93 (14.3) | |
| Hospital department, n (%): | | | 0.005 |
| clinical | 1266 (90.2) | 138 (9.8) | |
| non-clinical | 477 (85.8) | 79 (14.2) | |
| Working in COVID-19 area, n (%): | | | 0.534 |
| yes | 133 (90.5) | 14 (9.5) | |
| No | 1610 (88.8) | 203 (11.2) | |
| Wards, n (%): | | | 0.040 |
| medical | 734 (90.1) | 81 (9.9) | |
| surgical | 151 (92.6) | 12 (7.4) | |
| intensive care | 81 (86.2) | 13 (13.8) | |
| auxiliary | 206 (92.4) | 17 (7.6) | |
| ambulatory | 94 (86.2) | 15 (13.8) | |
| laboratory | 116 (87.9) | 16 (12.1) | |
| maintenance | 71 (82.6) | 15 (17.4) | |
| administration | 258 (86.6) | 40 (13.4) | |
| Other | 32 (80.0) | 8 (20.0) | |
| SARS-CoV-2 serostatus prior to the study, n (%) [c]: | | | 0.066 |
| positive | 189 (87.5) | 27 (12.5) | |
| Negative | 443 (91.9) | 39 (8.1) | |
| Previous SARS-CoV-2 infection, n (%) [d]: | | | < 0.001 |
| yes | 285 (81.0) | 67 (19.0) | |
| No | 1 458 (90.7) | 150 (9.3) | |
| Time between previous SARS-CoV-2 infection and vaccination status; median (IQR), days[e] | 61 (44–71) | 20 (0–55) | < 0.001 |
| Previous symptomatic infection, n (%): | | | 0.123 |
| yes | 123 (77.4) | 36 (22.6) | |
| no | 161 (83.8) | 31 (16.2) | |

Abbreviation: OR–odds ratio.

[a] received at least one dose of the BNT162b2 vaccine between January 4, 2021, and February 28, 2021

[b] did not receive any dose of vaccine between January 4, 2021, and February 28, 2021

[c] SARS-CoV-2 serology (IgG anti-nucleocapsid antibodies) was known for 698 HCWs prior to phase 2 of the study

[d] at time frame before vaccine dose 1 receipt or March 1, 2021

[e] date of vaccination availability (January 4, 2021) was considered as the end date to calculate the median time between date of previous PCR positive test result and vaccination status. A time value of 0 was assigned for 24 HCWs who encountered SARS-CoV-2 infection between January 4, 2021, and March 1, 2021, or the date of vaccine dose 1 receipt, whichever occurred earlier

bundle of intensified infection control measures) reduced the number of SARS-CoV-2 infections by early identification and isolation of SARS-CoV-2–positive individuals.

In line with some other studies conducted after the first wave of the pandemic, the changes in the incidence rate of SARS-CoV-2 infection among HCWs in our cohort were closely

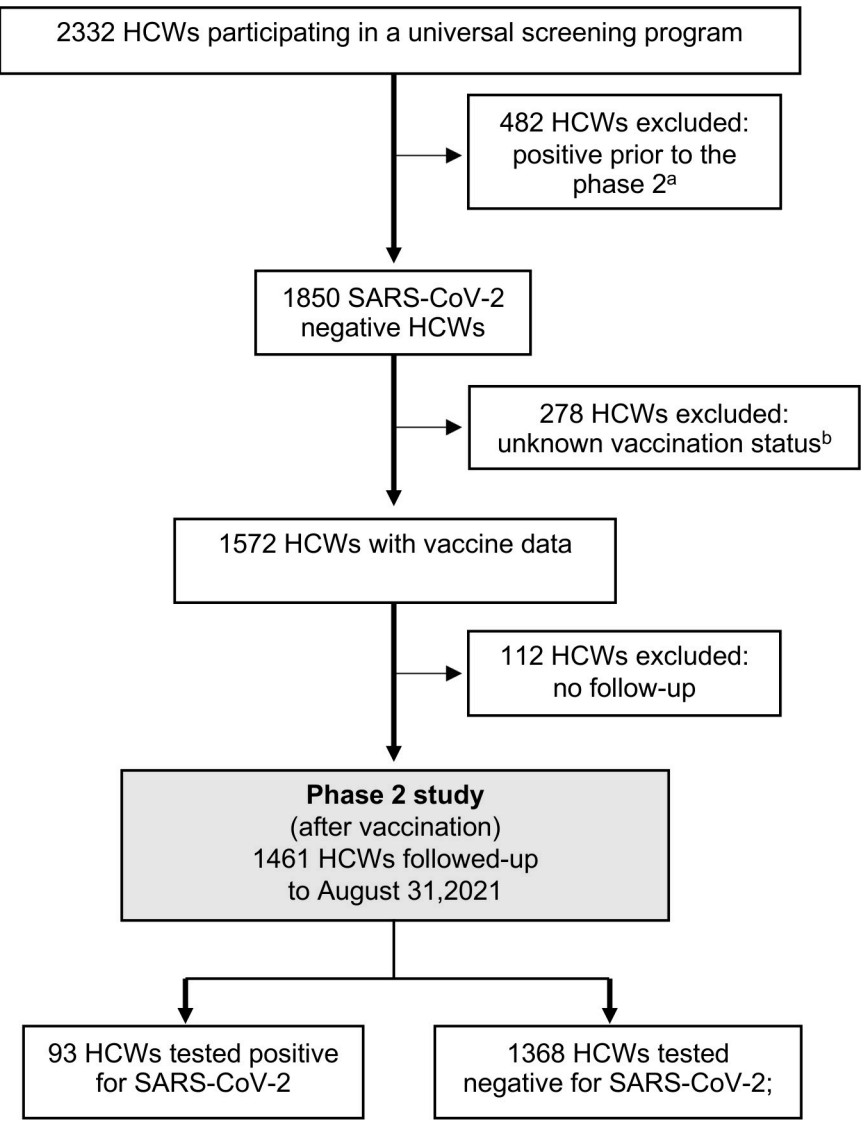

**Fig 4. Flow diagram of the participant inclusion and exclusion criteria used to define study cohort in phase 2 of the study (note that HCWs may meet more than one exclusion criterion).** [a] including 436 with SARS-CoV-2 PCR confirmed and 46 seropositive prior to phase 2; 4 out of 482 HCWs experienced reinfection (n = 2) or breakthrough (n = 2, fully vaccinated) infection within phase 2. [b] none of the HCWs with unknown vaccination status had SARS-CoV-2 infection detected.

followed by community infection rates [10, 18, 19]. Especially in the period before vaccination (study phase 1), both the dynamic and the magnitude of new SARS-CoV-2 infections were similar. On the other hand, after deployment of the HCW vaccination programme, a reduction in the weekly incidence of new cases among HCWs was observed. This downward trend initially was in parallel with the incidence decline in the community but persisted up to epi week 11, 2021, while in the general population a rapid increase in SARS-CoV-2 incidence was observed attributed to the SARS-CoV-2 Alpha variant domination (corresponding to the third pandemic wave). The incidence rate among HCWs also increased afterwards, but the observed peak was relatively lower. This finding is consistent with the anticipated protective effect of the COVID-19 vaccine.

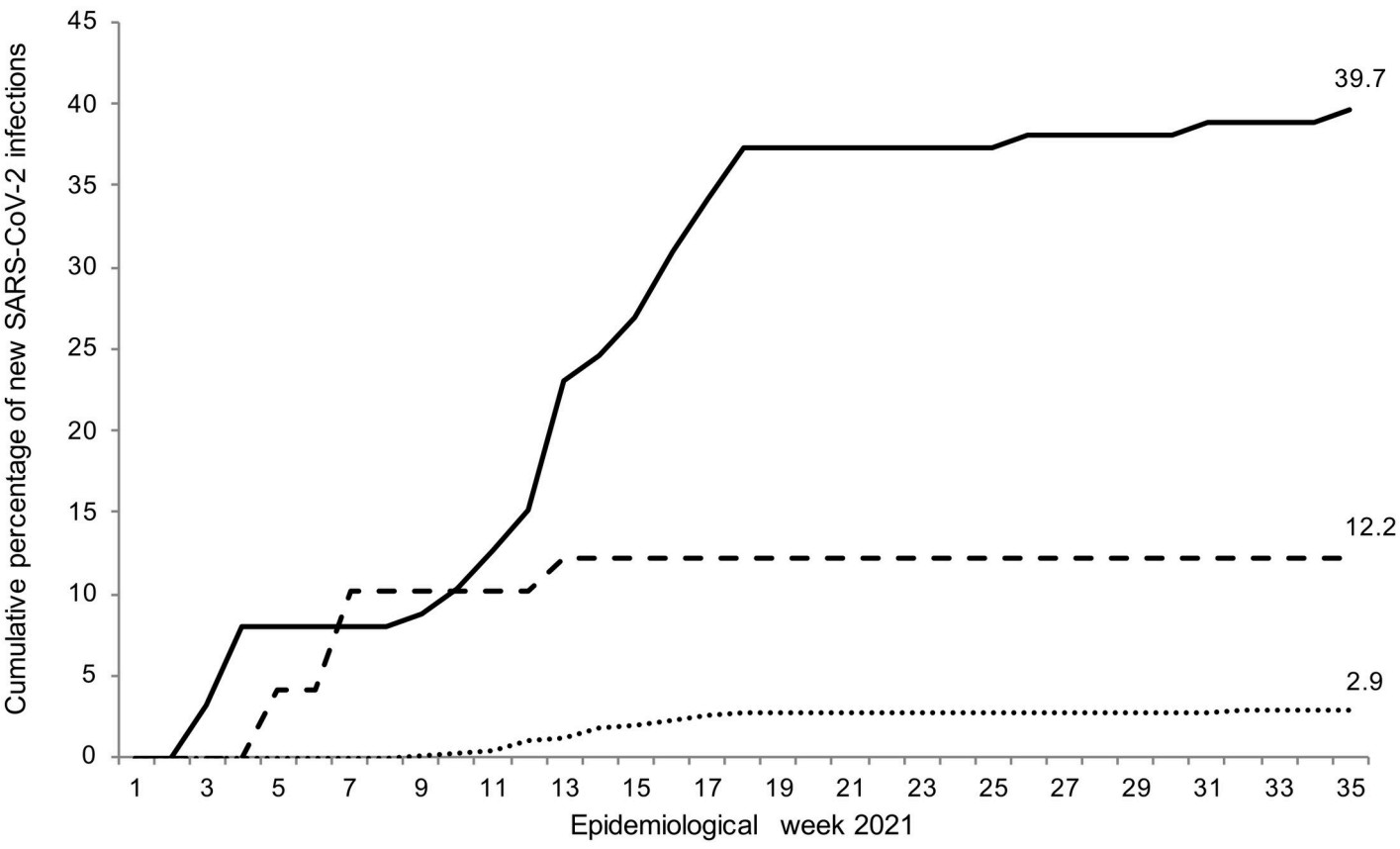

**Fig 5. Cumulative percentage of new SARS-CoV-2 infections in phase 2 of the study by epidemiological week.**

When considering the two phases of the present study, the cumulative incidence of new SARS-CoV-2 infections decreased from 16.6% before vaccination to 6.3% after the implementation of the vaccination programme at the CMHI (study phase 2). Due to the observational design of our study, we were unable to assess the extent to which HCW vaccination contributed to the observed reduction in the incidence of HCW SARS-CoV-2 infections. However, as the uptake of vaccination among HCWs at the CMHI was high (88.9%), we hypothesise that this impact could be significant, especially when the short-term effect is considered. In addition, this hypothesis was supported by the high vaccine effectiveness observed in our cohort. Our findings indicate that the effectiveness of the vaccine was 95% and 79% in fully and partially vaccinated HCWs, respectively. These results are comparable to previous reports including those from clinical trials and other real-world studies [20–22]. In line with a previous study conducted among HCWs, we did not observe significant differences in vaccine effectiveness by age or occupational groups [20, 22], although we observed slightly lower vaccine efficacy among HCWs in the 'other without direct patient contact' group. This finding could reflect lower awareness, behavior change, and a misbelief that vaccination allows infection control measures to be relaxed. However, despite high vaccine efficacy and high vaccination coverage, we still observed new SARS-CoV-2 infections among HCWs at a rate from 2.9% in the fully vaccinated to 39.7% in those unvaccinated. These numbers are higher than those reported

**Table 5. Vaccine effectiveness (VE) against any SARS-CoV-2 infection by age group and professional category.**

| VE by professional category: | Total, n | Infected, n | Unadjusted | Adjusted |
|---|---|---|---|---|
| | | | VE% (95% CI) | VE% (95% CI) |
| **Age group[a]:** | | | | |
| **<50 years** | 771 | 59 | | |
| fully vaccinated | 691 | 24 | 95 (92–97) | 95 (90–97) |
| **50+ years** | 641 | 28 | | |
| fully vaccinated | 595 | 13 | 95 (89–98) | 95 (87–98) |
| **Professional category[b]:** | | | | |
| **Nurse** | 421 | 27 | | |
| fully vaccinated | 384 | 9 | 97 (94–99) | 98 (92–99) |
| **Physician** | 327 | 15 | | |
| fully vaccinated | 317 | 10 | 97 (87–99) | 96 (84–99) |
| **other with direct patient contact[c]** | 177 | 11 | | |
| fully vaccinated | 156 | 1 | - | - |
| **other without direct patient contact** | 487 | 34 | | |
| fully vaccinated | 429 | 17 | 90 (79–95) | 89 (73–96) |

Abbreviations: VE–vaccine effectiveness; CI–confidence interval

[a] adjusted for gender and no. of tests per person (adjustment variables excluded age)

[b] adjusted for age, gender and number of tests per person

[c] VE for 'other with direct patient contact' category was not estimated because there was a small number of cases among fully vaccinated HCWs

previously. In a systematic review of eighteen studies, the pooled proportion of SARS-CoV-2 infection was 1.3%, 2.3%, and 10.1% among fully vaccinated, partially vaccinated, and unvaccinated HCWs, respectively [15]. Variability in demographic and occupational factors, as well as case definitions, could explain this difference. Furthermore, the fact that we excluded HCWs with a previous SARS-CoV-2 infection from our analysis may have affected the higher infection rates observed in our cohort (the absence of the protective effect of post-infection immunity). Although rare, these infections might diminish HCWs' belief in vaccination effectiveness, especially in those becoming infected with SARS-CoV-2 after initial vaccination. It is worth noting that, at the time of manuscript preparation, we faced a low rate (approximately 5%) of second booster dose uptake among HCWs at the CMHI. This led to concerns when considering waning vaccine immunity over time and the emergence of new variants of concern (VOC).

A systematic review by Biswas et al. revealed that vaccine acceptance varied widely between countries and ranged from 4.3% to 72% [23]. Individuals who were men, older, physicians, or well educated had a lower hesitancy to receive the COVID-19 vaccine. Other factors associated with higher vaccine acceptance were: higher income, medical risk, chronic disease history, not being infected with SARS-CoV-2 in the past, knowledge of COVID-19, and a belief that vaccines may protect friends, family, and community members [23]. Although the present study was not designed to assess the attitude to vaccination, these findings are consistent with our study. We observed that older individuals, physicians, those working in clinical settings, and those previously uninfected were more likely to be vaccinated. These observations might inform tailored communication strategies to be implemented to increase the uptake rate of COVID-19 vaccines among HCWs.

Reliable data on risk factors, SARS-CoV-2 incidence, and the proportion of HCWs who remain naïve (had no history of SARS-CoV-2 infection and/or unvaccinated) are crucial to inform infection control strategies. Previous studies reported widely varying estimates of

incidence and risk factors for infection among HCW [24], and only a few studies assessed HCW infection and risk longitudinally [19]. When considering potential risk factors of SARS-CoV-2 infection among HCWs in our study, we did not find a significant difference in the infection rates among HCWs working in wards involved and not involved in the treatment of COVID-19 patients, clinical or non-clinical areas, or across different wards and these findings applied to both phases of the study. Our findings are in contrast to some previous studies, especially those studies performed during the early pandemic stage [25], which reported higher infection rates among HCWs working in COVID-19 wards with direct patient contact, but our findings are in line with the emerging literature pointing out that, except for breaches in PPE, the main risks to HCWs come from outside of work factors (in the community and household) [19, 26–29]. In our cohort, before implementation of the vaccination programme (study phase 1), nurses had the highest adjusted likelihood of being infected, while physicians had a lower likelihood. Surprisingly, in the post-vaccination study period, we did not find any association between risk of SARS-CoV-2 infection and professional category. The only factor significantly associated with SARS-CoV-2 infection in the multivariate analysis was vaccination status. The increased risk of SARS-CoV-2 infection for nurses was previously reported in many observational studies [30–32]. The higher risk among nursing staff has been consistently explained by their more frequent contact with and longer contact times with COVID-19 patients when compared with physicians [31, 33, 34]. However, in our study we did not observe an increase in the number of cases on the wards involved in the treatment of patients with COVID-19. Furthermore, only a small percentage of HCWs (with available data on the possible source of exposure) reported exposure at work, and even fewer had high-risk contact with a COVID-19 patient. A recent case-control study from Ireland, investigating the impact of demographic and work-related factors on the risk of SARS-CoV-2 infection after in-work exposure to a confirmed case of COVID-19, revealed that male sex, Eastern European nationality, exposure location, PPE use, and vaccination status all impact the likelihood of SARS-CoV-2 infection throughout the first, second, and third pandemic waves [35]. In this study, no individual job role was determined to have a consistently higher risk of infection after documented nosocomial exposure. It is likely that the increased risk of SARS-CoV-2 infection among nurses observed in our and other studies may be an indication of the cumulative risk of certain nursing staff roles, which have an increased intensity of contact with patients and other HCWs over time, and this increases the probability of infection. In addition, nurses make up a significant proportion of the total hospital staff (approximately one-third in our cohort), thus resulting in over-representation, which may influence the interpretation of risk [35].

Together, these observations highlight the need for awareness of non-patient care exposure risk that contributes to infection in HCWs and should be considered in infection control measures (i.e. universal masking, reinforcement of hand hygiene, and distancing).

The limitations of our data should be borne in mind. First, the retrospective and observational design of the study is subject to missing and incomplete data and thus to unmeasured confounding bias. Second, although our study cohort was stratified by different occupational categories, there may still be different exposure risks within the categories. Third, potential confounders may be present that were unaccounted for in the regression analyses (such as comorbidities, contact time with COVID-19 cases, movement of HCWs between high- and low-risk wards, compliance with PPE, or household-related factors) due to data limitations, and therefore the results deserves caution interpretation and may not generalise to other settings with different characteristics. Fourth, the clinical data regarding symptoms for each HCW were retrieved from an unstandardized database generated by the ICP team for epidemiological purposes and were not consistently available. In addition, we relied on self-reported symptoms and community/household exposures. Exposures in the workplace were revealed

through epidemiological investigations, including contact tracing, but are still subject to recall bias. Data on exposure was missing for almost 40% of the HCWs, therefore it was unknown whether SARS-CoV-2 PCR positive cases resulted from infections in the workplace or were acquired in the community or household. On the other hand, the exact source and direction of infection may only be inferred from epidemiological data combined with viral genomic data, otherwise assessing the source is often subjective. Fifth, the lack of detailed information on SARS-CoV-2 exposure and the use of PPE, prevented us from analysing the association of adherence to PPE over time and across different occupational categories with the risk of infection. Six, we attempted to include only new cases of SARS-CoV-2 infection, excluding HCWs with records of a positive test by PCR or anti-nucleocapsid IgG. However, as systematic testing by PCR was not performed before the study period and not all HCWs had participated in a previous serology screening programme, we cannot rule out that we included some previous positive cases in our incidence analysis. Seventh, testing in study phase 2 was less dense as the universal screening programme ended on May 6, 2021, thus HCWs with asymptomatic SARS-CoV-2 infection might be underrepresented. Eight, although our estimated vaccine effectiveness aligns with the results provided in other reports, this estimate is based on a relatively short follow-up period. Finally, our study took place prior to the emergence of the Delta variant of SARS-CoV-2, thus it might not be generalizable to an epidemic situation with a domination of another VOC, however, the evidence from the early period of the pandemic may inform the infection control policy in a vulnerable population with regard to other respiratory viruses.

To our knowledge, this is the only study of SARS-CoV-2 infection among HCWs conducted in the central-east region of Poland and one of the few studies based on robust data obtained mainly from proactive universal screening by PCR testing. The size and diversity of the cohort (all HCWs, including nurses, physicians, and other personnel providing direct and indirect patient care, as well as non-clinical staff) together with longitudinal data collected in the period before and after vaccination implementation in HCWs, meant that it was possible to examine risk factors of the new SARS-CoV-2 infection by demographic, occupational, and vaccination status (which was determined based on vaccination records instead of self-reporting, to mitigate recall bias).

In summary, our data adds new knowledge regarding the incidence of SARS-CoV-2 infection among HCWs at the largest paediatric tertiary hospital in Poland, in the pre- and post-vaccination periods. We found that after the implementation of vaccination, the risk of infection among HCWs remained relatively high despite the high coverage of vaccination and the high effectiveness of the vaccine. Although rare, breakthrough infections are challenging, as they may pose a risk to the vulnerable population. Furthermore, there are concerns about the decrease in vaccine effectiveness over time and during the emergence of new VOCs. Thus, continued efforts to promote infection control measures and vaccination (including booster vaccine doses) and distancing remain necessary during the ongoing COVID-19 pandemic.

## Supporting information

**S1 File. Supplementary methods.**
(DOCX)

**S1 Fig. New weekly cases of SARS-CoV-2 infections by testing mode among HCWs at the CMHI and in the Mazovian voivodeship.**
(TIF)

**S2 Fig. Distribution of the SARS-CoV-2 variants in Poland during the study period (data source: ECDC.** Data on SARS-CoV-2 variants in the EU/EEA. https://www.ecdc.europa.eu/en/publications-data/data-virus-variants-covid-19-eueea).
(TIF)

**S1 Table. Characteristics of the HCWs by SARS-CoV-2 infection status including 86 previously infected HCWs (sensitivity analysis, n = 2233).**
(DOCX)

**S2 Table. Association of demographic and occupational characteristics of the HCWs with SARS-CoV-2 infection before vaccination (study phase 1).**
(DOCX)

**S3 Table. Characteristics of vaccinated and unvaccinated HCWs (n = 1461).**
(DOCX)

**S4 Table. Characteristics of the HCWs stratified by SARS-CoV-2 infection status during study phase 2 (postvaccination; n = 1461).**
(DOCX)

**S5 Table. Association of vaccination status with primary laboratory confirmed SARS--CoV-2 infection, after inclusion of additional category of HCWs with missing vaccination status.**
(DOCX)

**S6 Table. Association of demographic and occupational characteristics of HCWs with SARS-CoV-2 infection after vaccination (study phase 2).**
(DOCX)

## Author Contributions

**Conceptualization:** Beata Kasztelewicz, Marek Migdał, Katarzyna Dzierżanowska-Fangrat.

**Data curation:** Beata Kasztelewicz, Marek Migdał, Katarzyna Dzierżanowska-Fangrat.

**Formal analysis:** Beata Kasztelewicz.

**Investigation:** Beata Kasztelewicz, Katarzyna Skrok, Julia Burzyńska.

**Methodology:** Beata Kasztelewicz, Julia Burzyńska.

**Resources:** Marek Migdał.

**Supervision:** Marek Migdał, Katarzyna Dzierżanowska-Fangrat.

**Visualization:** Katarzyna Skrok.

**Writing – original draft:** Beata Kasztelewicz, Katarzyna Skrok, Julia Burzyńska.

**Writing – review & editing:** Beata Kasztelewicz, Katarzyna Skrok, Julia Burzyńska, Marek Migdał, Katarzyna Dzierżanowska-Fangrat.

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
