## [Decision Letter · Decision Letter 0]

23 Jan 2024

PONE-D-23-40084Incidence of SARS-CoV-2 infection among healthcare workers before and after COVID-19 vaccination in a tertiary paediatric hospital in Warsaw: a retrospective cohort studyPLOS ONE

Dear Dr. Kasztelewicz,

Thank you for submitting your manuscript to PLOS ONE. After careful consideration, we feel that it has merit but does not fully meet PLOS ONE’s publication criteria as it currently stands. Therefore, we invite you to submit a revised version of the manuscript that addresses the points raised during the review process.

**ACADEMIC EDITOR: Please attend to all comments made by Reviewers and make necessary review and/or reconstructions where needed.**

We look forward to receiving your revised manuscript.

Kind regards,

Olatunji Matthew Kolawole, Ph.D.

Academic Editor

PLOS ONE

Additional Editor Comments:

The article require some review and re-construction as indicated by reviewers

Reviewers' comments:

Reviewer's Responses to Questions

**Comments to the Author**

1. Is the manuscript technically sound, and do the data support the conclusions?

Reviewer #1: Partly

Reviewer #2: Partly

Reviewer #3: Yes

2. Has the statistical analysis been performed appropriately and rigorously? 

Reviewer #1: No

Reviewer #2: Yes

Reviewer #3: Yes

3. Have the authors made all data underlying the findings in their manuscript fully available?

Reviewer #1: Yes

Reviewer #2: Yes

Reviewer #3: Yes

4. Is the manuscript presented in an intelligible fashion and written in standard English?

Reviewer #1: No

Reviewer #2: Yes

Reviewer #3: Yes

5. Review Comments to the Author

Reviewer #1: Kasztelewicz et al. presented the data for healthcare workers during COVID-19. Authors need to present data tables and figures in the correct order. The results and discussion are very vast and need to be very precise and conclusive.

Reviewer #2: This study investigates the Incidence of SARS-CoV-2 infection among HCWs before and after COVID-19 vaccination in Warsaw.

My comments are as below:

- First of all, the introduction has to be re-organized and clearly illustrated to readers regarding of incidence of SARS-CoV-2 infection not only among healthcare workers but also among normal populations before and after COVID-19 vaccination.

- What is the novelty of this work? First time in the world? (I don’t think so!). In the region?

- The tables of this work are informative, however the quality of 2provided figures are very low, therefore the authors should provide high quality figures.

- I would suggest adding a general conclusion Figure according to your own result (like a graphical abstract)

- The possibility of confounding in the results should be discussed.

Reviewer #3: The manuscript presents a scientifically accurate evaluation of a retrospective analysis of the incidence of Covid-19 infections in health care workers before and after the provision of the SARS Cov-2 vaccine. Due to the analysis of retrospective data and the associated limitations in the availability and quality of the data and thus variables, no surprisingly new findings are presented. However, the manuscript is convincing due to the meticulous scientific processing, the comprehensive explanation of the statistical methods used, the transparency of the data through the extensive supplement and also through the critical, balanced discussion of the weaknesses of the author's own data and evaluation.

Even after a literature search, I came to the conclusion that "this is the only study of the SARS-CoV-2 infection among HCWs in Poland and one of the few studies based on robust data obtained mainly from proactive universal screening PCR testing"

There was only one passage where I would change the wording, line 55 "small cohorts". These cohorts had between >1000 and <1500 subjects. In line 59 +60, the reader learns that "we retrospectively analyzed the large real-world testing database"60 large real-world testing database". 60 large real-world testing database" with 2332 HCWs. Please reflect the ratio.

The writing style is clear and the manuscript is well structured. This makes it very easy for the reader to follow.

Perhaps the tables could be presented more compactly in the printed version. In principle, nothing stands in the way of publication.

6. PLOS authors have the option to publish the peer review history of their article (what does this mean?). If published, this will include your full peer review and any attached files.

Reviewer #1: No

Reviewer #2: No

Reviewer #3: No

---

## [Author Response · Author response to Decision Letter 0]

5 Mar 2024

February 29, 2024

Olatunji Matthew Kolawole, PhD

Academic Editor

PLOS ONE 

Revision and resubmission of manuscript PONE-D-23-40084

Dear Dr. Olatunji Matthew Kolawole,

Thank you for the opportunity to revise our manuscript on “Incidence of SARS-CoV-2 infection among healthcare workers before and after COVID-19 vaccination in a tertiary paediatric hospital in Warsaw: a retrospective cohort study”.

We appreciate you and the reviewers for their time in reviewing our manuscript and providing valuable comments. The manuscript has been reviewed and adjusted to meet PLOS ONE’s style requirements, including those for file naming. The reference list has been thoroughly examined and updated (none of the references has been retracted). The figure files were uploaded to the PACE digital diagnostic tool. The figures are provided in tiff format with a resolution of 300 ppi. 

We have carefully addressed all the comments. Our point-by-point responses to the reviewers’ comments are provided below, and the online submission form will be updated accordingly.

All line numbers refer to the revised manuscript with tracked changes.

We appreciate your consideration and look forward to your response.

Sincerely,

Beata Kasztelewicz

Corresponding Author

Responses to the Reviewers’ comments on the manuscript:

“Incidence of SARS-CoV-2 infection among healthcare workers before and after COVID-19 vaccination in a tertiary paediatric hospital in Warsaw: a retrospective cohort study”. 

Reviewer #1:

Kasztelewicz et al. presented the data for healthcare workers during COVID-19. Authors need to present data tables and figures in the correct order.

Thank you for this suggestion. To address this concern and clarify the Result section we split Figure 3 (Study Flowchart) into two figures separate for Phase 1 (Fig 3) and Phase 2 (Fig 4) of the study. We decided to maintain the order of the tables as we believe it is appropriate.

The results and discussion are very vast and need to be very precise and conclusive.

Thank you for this suggestion. We have considered all the above comments. We went through the Results and Discussion sections. We have added two subsections with informative headings (“Characteristics of the Study Population” and “SARS-CoV-2 RNA PCR testing”) in the Results section (lines 221 and 256, respectively). We have shortened and simplified Tables 2 and 3 by removing 10 and 4 rows, respectively. In addition, the Discussion section was shortened by removing following passage on line 436:

“There are also some issues associated with massive asymptomatic testing that must be underlined. First, it requires additional operational and testing costs and a large laboratory capacity to provide rapid turnaround for testing. Second, the results of the PCR-based screening are valid only for the day of the test, which can cause a false sense of confidence[18]. Furthermore, the resources required to identify a single asymptomatic case are substantial and may not be cost-effective during a low prevalence period[19]. Thus, testing strategies should be implemented after careful consideration of resources, infrastructure capacity, and logistical issues[20]. In addition, the testing strategy must be guided by the local epidemiology, vaccination coverage, and efficacy among HCWs and in the community”.

We hope that this clearly and concisely sets the information.

Reviewer #2:

This study investigates the Incidence of SARS-CoV-2 infection among HCWs before and after COVID-19 vaccination in Warsaw.

My comments are as below:

- First of all, the introduction has to be re-organized and clearly illustrated to readers regarding of incidence of SARS-CoV-2 infection not only among healthcare workers but also among normal populations before and after COVID-19 vaccination.

Thank you for your suggestion. We have addressed this by removing the following passages from the Introduction section:

on line 39:

“…including universal masking, enhanced hand hygiene and personal protective equipment (PPE) training, symptom screening, and self-isolation of HCWs if symptomatic or in the case of close contact with an infected person”.

and on line 44:

“Following the implementation of COVID-19 vaccines, HCWs were among the first prioritised for vaccination in many countries, including Poland. Data from the placebo-controlled randomized phase 3 clinical trial of the Pfizer-BioNTech BNT162b2 vaccine showed 95% efficiency in preventing symptomatic SARS-CoV-2 infection [6]. Benefits similar to those observed in clinical trials were also observed in real-world conditions [7].”

We also provided information on the number of SARS-CoV-2 infections in the general population during the study period (before and after the introduction of COVID-19 vaccination), as suggested. In particular, the following passage on lines 45-54 has been added:

“In Poland, COVID-19 vaccination programme started on December 27, 2020, using the Pfizer-BioNTech BNT162b2 vaccine. HCWs were among the initial groups that prioritized for vaccination. The introduction of vaccination in Poland coincided with the descending phase of the second wave of the pandemic. At the end of December 2020, 1,289,293 COVID-19 cases were reported in Poland. Masovian Voivodeship was the region most affected by COVID-19, with Warsaw having the highest number of infections, with 64,813 confirmed cases by the end of 2020 [6,7]. Between February and April 2021, Masovian Voivodeship, like the rest of Poland, experienced a third wave of pandemics, with a peak of 5,264 daily cases reached on March 26, 2021. The number of cases decreased by June 2021, and by the end of August 2021, there were 45 daily cases of SARS-CoV-2 infections in Masovian Voivodeship [7,8].

We believe that this in the Introduction section provides an additional context to the reader.

Regarding infection prevalence among HCWs, there are no official reports available for Poland (which was one of the rationales for our study). In the Introduction section, we have added the following passage with the results of an early report by Papoutsi et al. (i.e., before April 17, 2020) on the prevalence of infections among HCWs in different regions of the world, including Poland - lines 55-71:

” An early report on the global burden of the COVID-19 pandemic on HCWs showed high variability in HCW infection percentage among total case across regions, with a median of 10.04% and range from 0 to 24.09%. Of note, Poland was one of the countries with high percentage of HCWs infections accounting for 17.07% of cases [9].”

Data on infections among HCWs in the European region later in the pandemic are limited, and we have highlighted this issue in the Introduction section.

- What is the novelty of this work? First time in the world? (I don’t think so!). In the region?

Thank you for pointing this out. We respond by adding the following passage to the Discussion on lines 584-585: “In summary, our data adds new knowledge regarding the incidence of SARS-CoV-2 infection among HCWs at the largest paediatric tertiary hospital in Poland, in the pre- and post-vaccination periods.”

In addition, a passage on lines 576-578 was rephrased to underline that “this is the only study of SARS-CoV-2 infection among HCWs conducted in the central-east region of Poland and one of the few studies based on robust data obtained mainly from proactive universal screening by PCR testing”.

- The tables of this work are informative, however the quality of 2 provided figures are very low, therefore the authors should provide high quality figures.

Thank you for pointing this out. We have provided high-quality figures as suggested. All figures were verified using the PACE digital diagnostic tool. The figures are provided in tiff format, with resolution of 300 ppi.

- I would suggest adding a general conclusion Figure according to your own result (like a graphical abstract)

Thank you for the suggestion. We have prepared a graphical abstract for our study, which will be submitted as a PDF file.

- The possibility of confounding in the results should be discussed.

Thank you for pointing this out. We addressed confounding in the Discussion section describing the limitations of the study. To highlight this, we rephrase the passage on lines 546-552 as follows “Third, potential confounders may be present that were unaccounted for in the regression analyses (such as comorbidities, contact time with COVID-19 cases, movement of HCWs between high- and low-risk wards, compliance with PPE, or household-related factors) due to data limitations, and therefore the results deserves caution interpretation and may not generalise to other settings with different characteristics.”

Reviewer #3:

The manuscript presents a scientifically accurate evaluation of a retrospective analysis of the incidence of Covid-19 infections in health care workers before and after the provision of the SARS Cov-2 vaccine. Due to the analysis of retrospective data and the associated limitations in the availability and quality of the data and thus variables, no surprisingly new findings are presented. However, the manuscript is convincing due to the meticulous scientific processing, the comprehensive explanation of the statistical methods used, the transparency of the data through the extensive supplement and also through the critical, balanced discussion of the weaknesses of the author's own data and evaluation.

Even after a literature search, I came to the conclusion that "this is the only study of the SARS-CoV-2 infection among HCWs in Poland and one of the few studies based on robust data obtained mainly from proactive universal screening PCR testing"

We would like to thank the reviewer for the considerable time, effort, and suggestions for improving the quality of this paper.

There was only one passage where I would change the wording, line 55 "small cohorts". These cohorts had between >1000 and <1500 subjects. In line 59 +60, the reader learns that "we retrospectively analyzed the large real-world testing database"60 large real-world testing database". 60 large real-world testing database" with 2332 HCWs. Please reflect the ratio.

Thank you for pointing this out. We agree that the statements in the paragraph have been misleading. We have revised the paragraph and removed “in small cohorts of HCWs on line 75 and “large” on line 78.

The writing style is clear and the manuscript is well structured. This makes it very easy for the reader to follow. Perhaps the tables could be presented more compactly in the printed version. In principle, nothing stands in the way of publication.

Thank you again. The tables have been simplified, as suggested. In particular, Tables 2 and 3 were shortened by removing 10 and 4 rows, respectively. We believe this makes them more concise.

---

## [Editor Report · Decision Letter 1]

20 Mar 2024

Incidence of SARS-CoV-2 infection among healthcare workers before and after COVID-19 vaccination in a tertiary paediatric hospital in Warsaw: a retrospective cohort study

PONE-D-23-40084R1

Dear Dr. Kasztelewicz,

We’re pleased to inform you that your manuscript has been judged scientifically suitable for publication and will be formally accepted for publication once it meets all outstanding technical requirements.

An invoice for payment will follow shortly after the formal acceptance. To ensure an efficient process, please log into Editorial Manager at Editorial Manager® , click the 'Update My Information' link at the top of the page, and double check that your user information is up-to-date. If you have any billing related questions, please contact our Author Billing department directly at authorbilling@plos.org.

Kind regards,

Olatunji Matthew Kolawole, Ph.D.

Academic Editor

PLOS ONE
---

## [Editor Report · Acceptance letter]

13 May 2024

PONE-D-23-40084R1 

PLOS ONE

Dear Dr. Kasztelewicz, 

I'm pleased to inform you that your manuscript has been deemed suitable for publication in PLOS ONE. Congratulations! Your manuscript is now being handed over to our production team.

Kind regards, 

on behalf of

Dr. Olatunji Matthew Kolawole 

Academic Editor

PLOS ONE